# Endophytic *Colletotrichum* Species from Aquatic Plants in Southwest China

**DOI:** 10.3390/jof8010087

**Published:** 2022-01-16

**Authors:** Hua Zheng, Zefen Yu, Xinwei Jiang, Linlin Fang, Min Qiao

**Affiliations:** 1Laboratory for Conservation and Utilization of Bio-Resources, Key Laboratory for Microbial Resources of the Ministry of Education, Yunnan University, Kunming 650091, China; zhenghua@mail.ynu.edu.cn (H.Z.); zfyu@ynu.edu.cn (Z.Y.); jiangxinwei@mail.ynu.edu.cn (X.J.); fllcsz@mail.ynu.edu.cn (L.F.); 2School of Life Sciences, Yunnan University, Kunming 650091, China

**Keywords:** morphology, multi-locus, novel species, pathogenicity, taxonomy

## Abstract

*Colletotrichum* species are plant pathogens, saprobes, and endophytes in many economically important hosts. Many studies have investigated the diversity and pathogenicity of *Colletotrichum* species in common ornamentals, fruits, and vegetables. However, *Colletotrichum* species occurring in aquatic plants are not well known. During the investigation of the diversity of endophytic fungi in aquatic plants in southwest China, 66 *Colletotrichum* isolates were obtained from aquatic plants there, and 26 of them were selected for sequencing and analyses of actin (ACT), chitin synthase (CHS-1), glyceraldehyde-3-phosphate dehydrogenase (GAPDH), the internal transcribed spacer (ITS) region, and β-tubulin (TUB2) genomic regions. Based on morphological characterization and multi-locus phylogenetic analyses, 13 *Colletotrichum* species were recognized, namely, *C. baiyuense* sp. nov., *C. casaense* sp. nov., *C. demersi* sp. nov., *C. dianense* sp. nov., *C. fructicola*, *C. garzense* sp. nov., *C. jiangxiense*, *C. karstii*, *C. philoxeroidis* sp. nov., *C. spicati* sp. nov., *C. tengchongense* sp. nov., *C. vulgaris* sp. nov., *C. wuxuhaiense* sp. nov. Two species complexes, the *C. boninense* species complex and *C. gloeosporioides* species complex, were found to be associated with aquatic plants. Pathogenicity tests revealed a broad diversity in pathogenicity and aggressiveness among the eight new *Colletotrichum* species.

## 1. Introduction

Aquatic plants are the plant groups that are physiologically attached to the aquatic environment, and at least part of their life cycle occurs in the water or on the surface of the water [1]. Usually, they were classified into five types according to their growth forms: emergent plants, floating-leaved plants, free-floating plants, submerged plants, and wet plants. As an important part of the structure and function of aquatic ecosystems, aquatic plants are a key group in maintaining the healthy operation of aquatic ecosystems, and they play an important role in them [2,3,4]. Their functions are mainly reflected in maintaining water quality, removing excessive nutrient load, absorbing nutrient mineral ions, and reducing sediment resuspension [3,5,6].

Fungal endophytes refer to kinds of fungi that live in the tissues and organs of healthy plants at a certain stage or all stages of a plant’s life cycle and cause no symptomatic infections [7]. These fungi are a significant part of the plant microbiome. A previous study found that fungal communities associated with plants are not randomly assembled [8]; plants tend to recruit a particular microbial consortium to adapt themselves to the environmental conditions [9,10]. Endophytes play specific roles for plants by interacting with hosts to perform a range of associations and functions from defensive mutualism and enhancement of stress tolerance to latent pathogenicity [11,12,13,14]. In addition, many studies have reported that endophytes frequently produce diverse secondary metabolites, many of which are vital in agriculture, industry, and medicine [15,16,17,18]. Therefore, fungal endophytes also serve as important sources of new and novel metabolites that are bioactive [19]. As of now, endophytes have been investigated in many floras, such as seagrasses [20], lichens [21], herbs and trees, etc. [22,23,24]. However, the vast majority of studies of endophytes focused on plants in terrestrial systems [25,26,27,28,29], while plants in marine and freshwater ecosystems have received little attention.

The genus *Colletotrichum* was established by Corda (1831) with *C. lineola* as the type species and is the sole genus in the family Glomerellaceae (Glomerellales, Sordariomycetes, Ascomycota) [30,31,32,33,34]. *Colletotrichum* species are globally distributed and are important plant and human pathogens [35,36,37,38,39,40], endophytes [41,42,43,44], and saprobes [43,45,46]. However, the taxonomy of *Colletotrichum* species has proven to be challenging. *Colletotrichum* species are mostly similar, which makes it difficult to identify species solely based on their morphology [47]. Previous research also investigated *Colletotrichum* species as pathogens associated with common fruits based on their morphology and ITS sequence data [48,49], but it did not distinguish some closely related taxa in several species complexes [50]. Therefore, a polyphasic approach based on morphology and genetic characteristics was proposed for the identification of *Colletotrichum* species [51]. For instance, a combination of multiple gene sequences, including the internal transcribed spacer (ITS), glyceraldehyde 3-phosphate dehydrogenase (GAPDH), chitin synthase (CHS-1), actin (ACT), and beta-tubulin (TUB2), can provide more molecular features to resolve different species in a *Colletotrichum* species complex [31,46,47]. Recently, several studies have analyzed this genus, and they have provided an account of 248 currently accepted species with molecular data, which fall into 14 different species complexes and 13 singleton species clades [46,47,52]. 

Southwest China is one of the world’s 34 biodiversity hotspots [53]. During these years, our fungal diversity research, which investigates aquatic plants [54,55,56], submerged leaves [57,58,59,60,61], and soils [62,63,64], confirmed that this area harbors an inestimable diversity of fungi. In our previous study, we investigated the diversity of endophytic fungi in aquatic plants in southwest China and obtained 1689 fungal isolates, including 66 strains belonging to *Colletotrichum* species based on the ITS sequence [65]. Then, we selected 26 of them for sequencing of other loci, including ACT, CHS-1, GAPDH, and TUB2. Finally, 13 *Colletotrichum* species were recognized based on a five-locus phylogenetic analysis and morphological characteristics, namely, *C. baiyuense* sp. nov., *C. casaense* sp. nov., *C. demersi* sp. nov., *C. dianense* sp. nov., *C. fructicola*, *C. garzense* sp. nov., *C. jiangxiense*, *C. karstii*, *C. philoxeroidis* sp. nov., *C. spicati* sp. nov., *C. tengchongense* sp. nov., *C. vulgaris* sp. nov., and *C. wuxuhaiense* sp. nov. Pathogenicity tests revealed that eight new species caused symptoms in artificially wounded fruit. This study expands the worldwide diversity of *Colletotrichum* species and provides descriptions of new taxa.

## 2. Materials and Methods

### 2.1. Collection of Aquatic Plant Samples

From 2014 to 2015, we sampled aquatic plants in Yunnan, Guizhou, and Sichuan Provinces, mainly from wetlands, reservoirs, ponds, lakes, and rivers. Detailed information on sampling sites and plant species can be found in Zheng et al. [65]. Healthy, mature plants with undamaged leaves were fully uprooted, the mud was washed off them, and then they were brought to the laboratory in sterile polythene bags. During transportation, plant samples were stored at 4 °C.

### 2.2. Isolation of Endophytic Fungi

Each plant sample was cut into segments of 20–30 mm in length and washed thoroughly with tap water. Then, plant samples were processed using the following sequence of steps: initial immersion for 2 min in 0.5% sodium hypochlorite, followed by 1 min in sterile distilled water, 2 min in 75% ethanol, and, finally, 1 min in sterile distilled water. The efficacy of the sterilization procedure was confirmed by the method of Schulz et al. [66]. After surface disinfection, all samples were cut further into smaller sections of about 5 mm × 5 mm × 5 mm, and were then evenly spaced in a 90 mm Petri dish containing Rose Bengal Agar (RBA; Guangdong Huankai Microbial Sci and Tech, Guangzhou, China). Two antibiotics (penicillin G (0.5 g/L) and streptomycin (0.5 g/L)) were added to suppress bacterial growth [67]. Petri dishes were sealed, incubated at 25 °C, and examined periodically. Fungal mycelia growing out from the plant tissues were transferred to other plates containing potato dextrose agar (PDA; potato 200 g, glucose 20 g, agar 18 g, 1 L distilled water).

The pure cultures and dried cultures were deposited in the Herbarium of the Laboratory for Conservation and Utilization of Bio-Resources, Yunnan University, Kunming, Yunnan, China (YMF) and the China General Microbiological Culture Collection Center (CGMCC).

### 2.3. Morphological Characterization

Agar plugs (6 mm in diameter) were taken from the edge of actively growing cultures on PDA and transferred to the center of 9-cm-diameter Petri dishes of PDA, cornmeal dextrose agar (CMA; 20 g cornmeal, 18 g agar, 1 L distilled water), oatmeal agar (OA; 40 g oatmeal, 18 g agar, 1 L distilled water), and synthetic low-nutrient agar (SNA; 1 g KH_2_PO_4_, 1 g KNO_3_, 0.5 g MgSO_4_, 0.5 g KCl, 0.2 g glucose, 0.2 g sucrose, 18 g agar, 1 L distilled water). Cultures were incubated at 25 °C with alternating 12 h of light and 12 h of darkness for 7 d. The colony characteristics, colony diameters, and pigment production on four media were recorded. Microscopic characters were observed using a light microscope (Olympus BX51), and photographs were taken on an Olympus BX51 microscope under a differential interference contrast model and captured with an Olympus DP 10 digital camera using the Olympus DP controller (V.3, 1, 1208) software. Appressoria were produced from hyphae using a slide culture technique described by Cai et al. [51]. The average measure of each microscopic structure was calculated from 30 measurements.

### 2.4. DNA Extraction, PCR Amplification, and Sequencing

For each *Colletotrichum* spp. isolate, the mycelium was taken from a 7-day-old culture grown on PDA medium and transferred to 2 mL Eppendorf micro-centrifuge tubes. Total genomic DNA was extracted according to the procedures in Turner et al. [68]. Five genetic fragments of *Colletotrichum* strains, namely, ITS, GAPDH, CHS-1, ACT, and TUB2, were amplified by using the following primer pairs: GDF1 and GDR1 for GAPDH [69], ACT512F and ACT783R for ACT [70], ITS4 and ITS5 for ITS [71], and Btub2Fd and Btub4Rd for TUB2 [72]. Each 25 μL PCR reaction volume consisted of 12.5 μL T5 Super PCR Mix (containing Taq polymerase, dNTP, and Mg^2+^, Beijing TsingKe Biotech Co., Ltd., Beijing, China), 1 μL of forwarding primer (10 μM), 1 μL of reverse primer (10 μM), 1 μL of DNA template, 5 μL of PCR buffer, and 4.5 μL of sterile water. PCR reactions were run in an Eppendorf Mastercycler (Eppendorf, Hamburg, Germany) following the PCR thermal cycle programs described by Chung et al. [73].

Then, all PCR products were purified using a commercial kit (Bioteke Biotechnology) following the manufacturer’s instructions. Purified PCR products were sequenced in both the forward and reverse sense with a LI-COR 4000L automatic sequencer using a Thermo Sequenase-kit as described by Kindermann et al. [74]. Raw sequence chromatograms were manually examined and then aligned to produce a consensus sequence for each isolate using ClustalW in MEGA 6.06 [75]. The consensus sequences were deposited in the GenBank database at the National Center for Biotechnology Information (NCBI), and the accession numbers are listed in Appendix A.

### 2.5. Sequence Alignment and Phylogenetic Analysis

The phylogeny was constructed with sequences of ITS, GAPDH, CHS-1, ACT, and TUB2. Related sequences of reference isolates were downloaded from GenBank based on recent publications [46,47]; all sequences are listed in Appendix A. Firstly, each gene region was aligned with Clustal X 1.83 [76] using the default parameters. Then, the five aligned sequences were manually adjusted; gap adjustments were done, and ambiguous regions were also excluded and linked by using BioEdit v.7.0 [77]. Phylogenetic analyses were based on the maximum likelihood (ML) and Bayesian inference (BI) methods for the multi-locus dataset. 

For the ML analysis, the concatenated sequence dataset (Fasta file) was converted into a PHY file with CLUSTAL_X version 1.83 [76], and then the analysis was performed by using RAxML [78] with the GTR-GAMMA model. Maximum likelihood bootstrap proportions (MLBPs) were also computed by the RAxML program with 1000 replicates. For the BI analysis, the dataset was converted into a NEXUS file with MEGA6 [75]. Then, the Akaike information criterion (AIC) was implemented by using jModelTest version 2.0 [79] to select the best-fit models of nucleotide substitution. GTR + I + G + F model was selected for the BI analysis, which was conducted on MrBayes 3.1.2 [80]. A Markov Chain Monte Carlo (MCMC) algorithm was used to generate phylogenetic trees. Two runs with four chains (one cold and three heated) were executed simultaneously for 5,000,000 generations and sampled every 500 generations. Of the trees, the initial 25% of the generations were discarded as burn-in, and the remaining trees were used to calculate the Bayesian inference posterior probability (BIPP) values. When the average standard deviation of split frequencies was less than 0.01, the analyses were stopped. Phylogenetic trees were visualized by using TreeView 1.6.6 [81], and the final topology of the phylogenetic tree is shown in Figure 1, with an MLBP greater than 70% and BIPP greater than 90% indicated for the respective clades.

### 2.6. Pathogenicity Tests

Selected isolates (one isolate from each new *Colletotrichum* species) were used for pathogenicity and aggressiveness tests on five common fruits, including strawberry, grape, tangerine, tomato, and blueberry. Each isolate was incubated on CMA for 7–15 days at 25 °C to achieve spore suspension. Fresh fruits without visible diseases were used for the tests. These fruits were surface sterilized in 75% ethanol for 3 min and 1% sodium hypochlorite for 3 min, and then with three rinses in sterile distilled water.

After washing and air drying, these fruits (six of each fruit) were wounded by stabbing to a depth of 3 mm to form a 3-mm-diameter circle, and 6 μL of spore suspension (1 × 10^6^ spores/mL) was inoculated on the wounds. Sterile water was used as a control treatment. Inoculated fruits were placed in glass containers on top of moist paper and sealed. The containers were placed in a growth chamber and incubated at 20 °C with alternating 12 h of light and 12 h of darkness for 14 days. Symptom development of the fruits was checked daily and recorded for up to 14 days. 

Moreover, the virulence levels of our isolated *Colletotrichum* spp. strains were evaluated according to the lesion size and number of spores as follows: (a) level 0: no hyphae; (b) level 1: only mycelia observed; (c) level 2: lesion diameter smaller than 0.5 cm and no spores produced; (d) level 3: lesion diameter larger than 0.5 cm and no spores produced; (e) level 4: few spores produced at the inoculation point; (f) level 5: large numbers of spores produced and expanded outward; (g) level 6: spores covering the entire fruit.

## 3. Results

### 3.1. Multi-Locus Phylogeny

Our 26 selected representative isolates together with 141 reference species (Appendix A)—belonging to 14 species complexes and 13 singleton species—were subjected to multi-locus phylogenetic analyses with concatenated CHS-1, GAPDH, ACT, TUB2, and ITS sequences. The concatenated datasets from the five loci were analyzed with the two methods of ML and BI, and *Monilochaetes infuscans* CBS 86996 was used as the outgroup. As the topology of the Bayesian analysis of the concatenated dataset was nearly identical to that of the ML consensus tree, only the Bayesian tree is shown, with bootstrap (from the ML analysis) on the left and posterior probability (from the Bayesian analysis) on the right of the corresponding nodes (Figure 1). 

The phylogenetic tree showed that seven isolates fell into the *Colletotrichum gloeosporioides* species complex clade. Of these, two of them clustered with *C. fructicola,* showing 99% MLBP and 97% BIPP, and three of them clustered with *C. jiangxiense,* showing 86% MLBP and 100% BIPP. The other two isolates in this complex formed a solitary clade corresponding to two new species, which were, respectively, designated as *C. tengchongense* and *C. vulgaris*. Five isolates fell into the *C. boninense* species complex clade. Of these, one of them clustered with *C. karstii,* showing 100% MLBP and 93% BIPP, and the other four isolates formed two sole clades representing two new species, which were, respectively, designated as *C. spicati* (MLBP/BIBP = 100%/100%) and *C. wuxuhaiense* (MLBP/BIBP = 100%/100%). Twelve isolates clustered together near to the *C. graminicola-caudatum* species complex clade and formed four clades representing four new species, which were, respectively, designated as *C. baiyuense* (MLBP/BIBP = 99%/100%), *C. casaense* (MLBP/BIBP = 85%/100%), *C. demersi* (MLBP/BIBP = 100%/100%), and *C. garzense* (MLBP/BIBP = 100%/100%). One isolate clustered with the singleton species *C. nigrum,* showing 99% MLBP and 100% BIPP; then, the isolate was proposed as a new species combined with a morphology and was designated as *C. dianense*. One isolate formed a solitary clade that was near to the *C. magnum* species complex clade and C. orchidearum species complex clade; finally, it was proposed as a new species and designated as *C. philoxeroidis*.

### 3.2. Taxonomy

*Colletotrichum baiyuense* Z.F. Yu and H. Zheng, sp. nov. (Figure 2)

MycoBank Number: MB842284

Etymology: The species epithet refers to Baiyu county, which is the site of collection of the type strain.

Holotype: YMF 1.04941, isolated as an endophyte from the leaf of *Hippuris vulgaris*, Baiyu county, Tibetan Autonomous Prefecture of Garzê, Sichuan Province, China, August 2015, Z.F. Yu, preserved by lyophilization (a metabolically inactive state) in the State Key Laboratory for Conservation and Utilization of Bio-Resources in Yunnan. Ex-type culture: CGMCC3.18941.

Sexual morph not observed. Asexual morph on SNA. Vegetative hyphae 1–8 μm diam., hyaline, smooth-walled, septate, branched. Chlamydospores not observed. Conidiomata: acervular, setae and conidiophores formed from roundish brown cells. Setae: pale brown to dark brown, smooth-walled, 2–4 septate, 77–125 μm long, tip acute. Conidiophore: hyaline to pale brown, smooth-walled, septate, branched, cylindrical to clavate, to 50 μm long. Conidiogenous cells: hyaline to pale brown, smooth-walled, cylindrical to clavate, 13–28 μm × 4–5 μm, with a gelatinous coating, opening 1–2 μm diam. Conidia: smooth-walled, hyaline, curved, aseptate, containing 1–2 guttules, 13.4–24.6 μm × 3.4–4.6 μm. Appressoria: solitary, ellipsoidal to clavate, smooth-walled, light brown to dark brown, 6.0–15.5 μm × 4.2–7.1 µm.

Culture characteristics: Colonies on SNA: flat, inconspicuously circular, radial margin, surface: hyaline to buff, reverse: hyaline, without pigmentation, covered with abundant, minute aerial mycelium, reaching 63–68 mm diam. in 7 days. Colonies on PDA: rough and wrinkled, radial margin, surface: gray, reverse: salmon to umber, without pigmentation, covered with abundant, dense, floccose aerial mycelium, reaching 66–69 mm diam. in 7 days. Colonies on MEA: flat, entire margin, surface: saffron, reverse: orange, without pigmentation, sparse aerial mycelium, reaching 65–69 mm diam. in 7 days. Colonies on OA: rough, circular, entire margins, surface: saffron towards the inner ring, white towards the outer ring, reverse: pale gray, without pigmentation, covered with abundant, dense floccose aerial mycelium in the outer ring, lacking aerial mycelium in the outer ring, reaching 58–60 mm diam. in 7 days.

Additional specimens examined: China, Sichuan Province, Tibetan Autonomous Prefecture of Garzê, Baiyu county, isolated as an endophyte from the stem of *Hippuris vulgaris*, August 2015, Z.F. Yu, living culture BY64; China, Sichuan Province, Tibetan Autonomous Prefecture of Garzê, Baiyu county, isolated as an endophyte from the leaf of *Hippuris vulgaris*, August 2015, Z.F. Yu, living culture BY75.

Notes: Phylogenetically, our three isolates of *Colletotrichum baiyuense* formed a solitary clade, and this species was near to two recognized species, *C. casaense* and *C. garzense*. Morphologically, the three species all have curved conidia, but *C. baiyuense* is different from *C. casaense* and *C. garzense* in culture characteristics and the shape of appressoria.

*Colletotrichum casaense* Z.F. Yu and H. Zheng, sp. nov. (Figure 3)

MycoBank Number: MB842286

Etymology: The species epithet refers to Casa Lake, which is the site of collection of the type strain.

Holotype: YMF 1.04947, isolated as an endophyte from the stem of *Hippuris vulgaris*, Casa Lake, Luhuo county, Tibetan Autonomous Prefecture of Garzê, Sichuan Province, China, August 2015, Z.F. Yu, preserved by lyophilization (a metabolically inactive state) in State Key Laboratory for Conservation and Utilization of Bio-Resources in Yunnan. Ex-type culture: CGMCC3.18947.

Sexual morph not observed. Asexual morph on SNA. Vegetative hyphae: 1–8 μm diam., hyaline, smooth-walled, septate, branched. Chlamydospores not observed. Conidiophores formed directly on hyphae: smooth-walled, hyaline, branched, septate, cylindrical to clavate, 18–32 μm × 4.0–4.5 μm. Conidiogenous cells: hyaline, smooth-walled, integrated, terminal, cylindrical to clavate, phialidic with visible periclinal thickening at the apex, 11.6–16.6 μm × 4.5–6.2 μm. Conidia: hyaline, smooth-walled, aseptate, curved, sometimes straight, tapering towards both ends, containing 2 guttules, 14.4–21.0 μm × 3.4–4.7 μm. Appressoria: solitary, navicular, lobate, irregular in outline or undulate margin, one- or two-celled, light brown to dark brown, 5.3–20.3 μm × 3.9–9.9 µm.

Culture characteristics: Colonies on SNA: flat, entire margin, surface: translucent, reverse: translucent, without pigmentation, covered with abundant, minute, villiform aerial mycelium, reaching 65–68 mm diam. in 7 days. Colonies on PDA: flat, radial margin, surface: white, reverse: white in the margin and saffron in the center, without pigmentation, covered with abundant, dense, flocculent aerial mycelium, reaching 50–60 mm diam. in 7 days. Colonies on MEA: rough and crinkly, entire margin, surface: whitish, reverse: pale white, without pigmentation, lacking aerial mycelium, reaching 60–63 mm diam. in 7 days. Colonies on OA: circular, indistinct margin, surface: white, reverse: pale white, without pigmentation, covered with abundant, dense aerial mycelium in the center, lacking aerial mycelium in the margin, reaching 60–63 mm diam. in 7 days.

Additional specimens examined: China, Sichuan Province, Tibetan Autonomous Prefecture of Garzê, Luhuo county, Casa Lake, isolated as an endophyte from the stem of *Hippuris vulgaris*, August 2015, Z.F. Yu, living culture KS17; China, Sichuan Province, Tibetan Autonomous Prefecture of Garzê, Baiyu county, isolated as an endophyte from the leaf of *Hippuris vulgaris*, August 2015, Z.F. Yu, living culture BY97; China, Sichuan Province, Tibetan Autonomous Prefecture of Garzê, Litang county, isolated as an endophyte from the root of *Myriophyllum spicatum*, August 2015, Z.F. Yu, living culture LT2.

Notes: Four *Colletotrichum casaense* isolates are phylogenetically close to *C. baiyuense* and *C. garzense*. Morphologically, the three species also have similarity in their conidial shape. However, the conidiogenous cells of *C. casaense* are obviously shorter than those of *C. baiyuense* and *C. garzense*.

*Colletotrichum demersi* Z.F. Yu and H. Zheng, sp. nov. (Figure 4)

MycoBank Number: MB842288

Etymology: The species epithet is derived from the host plant *Ceratophyllum demersum*.

Holotype: YMF 1.04946, isolated as an endophyte from the leaf of *Ceratophyllum demersum*, Hongfeng Lake, Guiyang city, Guizhou Province, China, August 2015, Z.F. Yu, preserved by lyophilization (a metabolically inactive state) in the State Key Laboratory for Conservation and Utilization of Bio-Resources in Yunnan. Ex-type culture: CGMCC3.18946.

Sexual morph not observed. Asexual morph on SNA. Vegetative hyphae: 1–8 μm diam., hyaline, smooth-walled, septate, branched. Chlamydospores not observed. Conidiophores formed directly on hyphae: smooth-walled, hyaline, branched, septate, cylindrical to clavate, 18–32 µm × 4.0–4.5 μm. Conidiogenous cells: hyaline, smooth-walled, integrated, terminal, cylindrical to clavate, phialidic with visible periclinal thickening at the apex, 11.6–16.6 µm × 4.5–6.2 μm. Conidia: hyaline, smooth-walled, aseptate, cylindrical, the apex rounded and base acute, sometimes with prominent scars, straight or slightly curved, 12.9–20.9 µm × 4.8–6.9 μm. Appressoria: solitary, clavate to lobed, smooth-walled, light brown to dark brown, 6.1–16.2 µm × 4.6–9.1 µm.

Culture characteristics: Colonies on SNA: flat, entire margin, surface: hyaline, reverse: hyaline, without pigmentation, lacking aerial mycelium, reaching 68–70 mm diam. in 7 days. Colonies on PDA: entire margin, surface: white, reverse: pale white, without pigmentation, covered with abundant, loose, floccose aerial mycelium, reaching 55–64 mm diam. in 7 days. Colonies on MEA: rough and crinkly, entire margin, surface: whitish, reverse: pale white, without pigmentation, lacking aerial mycelium, reaching 63–65 mm diam. in 7 days. Colonies on OA: indistinct margin, surface: white, reverse: pale white, without pigmentation, covered with loose, flocculent aerial mycelium in the center, lacking aerial mycelium in the margin, reaching 63–65 mm diam. in 7 days.

Additional specimen examined: China, Guizhou Province, Guiyang city, Hongfeng Lake, isolated as an endophyte from the leaf of *Ceratophyllum demersum*, August 2015, Z.F. Yu, living culture HF18.

Notes: Phylogenetically, *Colletotrichum demersi* formed a sole clade near to our three species studied here, which are *C. casaense, C. baiyuense,* and *C. garzense*, and the graminicola-caudatum species complex clade. However, *C. demersi* is different from our isolated three species by having cylindrical conidia. Moreover, the species in the graminicola-caudatum complex have a filiform appendage at the apex of the conidium [82].

*Colletotrichum dianense* Z.F. Yu and H. Zheng, sp. nov. (Figure 5)

MycoBank Number: MB842289

Etymology: The species epithet refers to Dian Lake, which is the site of collection of the type strain.

Holotype: YMF 1.04943, isolated as an endophyte from the root of *Alternanthera philoxeroides*, Dian Lake, Kunming city, Yunnan Province, China, July 2014, Z.F. Yu, preserved by lyophilization (a metabolically inactive state) in the State Key Laboratory for Conservation and Utilization of Bio-Resources in Yunnan. Ex-type culture: CGMCC3.18943.

Sexual morph not observed. Asexual morph on SNA. Vegetative hyphae: 1–8 μm diam., smooth-walled, hyaline, branched, septate. Chlamydospores not observed. Conidiophores formed directly on hyphae: smooth-walled, hyaline, branched, septate, up to 61 µm long. Conidiogenous cells: hyaline, smooth-walled, variable in size and shape, clavate or cylindrical, 0.7–2.6 µm in length. Conidia: aseptate, hyaline, smooth-walled, long and elliptical to cylindrical, sometimes slightly curved, the apex round and the base slightly pointed, a little constricted in the middle, containing 2–4 guttules, 8.0–24.5 µm × 2.4–4.8 µm. Appressoria: solitary, elliptical to long elliptical, clavate or lobed, smooth-walled, light brown to dark brown, 6.9–23.5 µm × 3.8–12.0 µm. 

Culture characteristics: Colonies on SNA: flat, radial margin, surface: translucent to white, reverse: pale white, without pigmentation, covered with abundant, loose aerial mycelium, reaching 50–54 mm diam. in 7 days. Colonies on PDA: entire margin, surface: white, reverse: yellow, yellow pigmentation observed, covered with abundant, dense aerial mycelium, reaching 38–47 mm diam. in 7 days. Colonies on MEA: entire margin, surface: white, reverse: pale white, without pigmentation, covered with abundant, flocculent aerial mycelium, reaching 70 mm diam. in 7 days. Colonies on OA: indistinct margin, surface: white, reverse: pale white, without pigmentation, aerial mycelia loose, reaching 50–52 mm diam. in 7 days. 

Notes: *Colletotrichum dianense* is phylogenetically close to the singleton species *C. nigrum* Ellis and Halst. with good support. Nevertheless, *C. dianense* can be easily distinguished from *C. nigrum* by its significantly differently shaped conidia. For instance, the conidial shapes of *C. dianense* are long elliptical to cylindrical, sometimes slightly curved, the apex round and the base slightly pointed, and a little constricted in the middle, while the conidia of *C. nigrum* are cylindrical, with a subacute apex [83]. Moreover, the GADPH sequence of *C. dianense* has 98% identity with that of *C. nigrum*.

*Colletotrichum garzense* Z.F. Yu and H. Zheng, sp. nov. (Figure 6)

MycoBank Number: MB842290

Etymology: The species epithet refers to Garzê prefecture, which is the site of collection of the type strain.

Holotype: YMF 1.04948, isolated as an endophyte from the stem of *Hippuris vulgaris*, Litang county, Tibetan Autonomous Prefecture of Garzê, Sichuan Province, China, August 2015, Z.F. Yu, preserved by lyophilization (a metabolically inactive state) in the State Key Laboratory for Conservation and Utilization of Bio-Resources in Yunnan. Ex-type culture: CGMCC3.18948.

Sexual morph not observed. Asexual morph on SNA. Vegetative hyphae: 1–8 μm diam., hyaline, smooth-walled, septate, branched. Chlamydospores not observed. Conidiophore formed directly on hyphae: smooth-walled, hyaline, branched, septate, cylindrical to clavate, up to 66 μm long. Conidiogenous cells: hyaline, smooth-walled, integrated, terminal, cylindrical to clavate, 11.6–16.6 µm × 4.5–6.2 μm. Conidia: hyaline, smooth-walled, aseptate, curved, tapering towards both ends, sometimes the end is round, 15.1–19.0 (–48.4) µm × 3.5–4.6 μm. Appressoria: solitary, lobate, irregular in outline or undulate margin, light brown to dark brown, 4.4–16.3 µm × 3.6–7.1 µm.

Culture characteristics: Colonies on SNA: flat, entire margin, surface: translucent, reverse: translucent, without pigmentation, covered with abundant, minute, villiform aerial mycelium, reaching 65–68 mm diam. in 7 days. Colonies on PDA: rough, indistinctly circular, entire margin, surface: white, grayish sepia ring in the margin, reverse: gray olivaceous, without pigmentation, covered with abundant, dense aerial mycelium, reaching 62–70 mm diam. in 7 days. Colonies on MEA: rough and crinkly, entire margin, surface: whitish, reverse: pale white, without pigmentation, lacking aerial mycelium, reaching 50–64 mm diam. in 7 days. Colonies on OA: circular, indistinct margin, surface: white, reverse: pale white, without pigmentation, covered with abundant, dense aerial mycelium in the center, lacking aerial mycelium in the margin, reaching 60–64 mm diam. in 7 days.

Additional specimens examined: China, Sichuan Province, Tibetan Autonomous Prefecture of Garzê, Litang county, isolated as an endophyte from the root of *Myriophyllum spicatum*, August 2015, Z.F. Yu, living culture LT31; China, Sichuan Province, Tibetan Autonomous Prefecture of Garzê, Litang county, isolated as an endophyte from the root of *Myriophyllum spicatum*, August 2015, Z.F. Yu, living culture LT76.

Notes: In our phylogenetic tree, *Colletotrichum garzense* clustered with two of our recognized species, *C. casaense* and *C. baiyuense*. In morphology, *C. garzense* differs from them by having smaller appressoria of 4.4–16.3 µm × 3.6–7.1 µm.

*Colletotrichum philoxeroidis* Z.F. Yu and H. Zheng, sp. nov. (Figure 7)

MycoBank Number: MB842291

Etymology: The species epithet is derived from the host plant *Alternanthera philoxeroides*.

Holotype: YMF 1.04945, isolated as an endophyte from the root of *Alternanthera philoxeroides*, Fuxian Lake, Chengjiang city, Yunnan Province, China, July 2014, Z.F. Yu, preserved by lyophilization (a metabolically inactive state) in the State Key Laboratory for Conservation and Utilization of Bio-Resources in Yunnan. Ex-type culture: CGMCC3.18945.

Sexual morph not observed. Asexual morph on SNA. Vegetative hyphae: 1–8 μm diam., hyaline, smooth-walled, septate, branched. Chlamydospores not observed. Conidiomata: acervular, setae and conidiophores formed from roundish brown cells. Setae: pale brown to dark brown, straight or slightly curved, smooth-walled, 2–3 septate, 102–144 μm long, tip acute. Conidiophores: smooth-walled, hyaline to pale brown, branched, septate, cylindrical to clavate, up to 35 μm long. Conidiogenous cells: smooth-walled, hyaline to pale brown, monoblastic, holoblastic, integrated, terminal, cylindrical to elongate-ampulliform, 11.6–16.6 µm × 4.5–6.2 μm, with a gelatinous coating, opening 1–2 μm diameter. Conidia: hyaline, smooth-walled, aseptate, the apex and base rounded, slightly constricted in the middle, 11.6–16.6 µm × 4.5–6.2 μm. Appressoria: solitary, labed, smooth-walled, light brown to dark brown, 6.3–12.1 µm × 6.7–11.8 µm.

Culture characteristics: Colonies on SNA: flat, entire margin, surface: translucent, reverse: translucent, without pigmentation, lacking aerial mycelium, reaching 44–45 mm diam. in 7 days. Colonies on PDA: circular, entire margin, surface: white to grayish sepia, the reverse has a uniform concentric ring with a whitish outside and olivaceous black inside, olivaceous buff in the center, without pigmentation, covered with abundant, dense, floccose aerial mycelium, reaching 68–70 mm diam. in 7 days. Colonies on MEA: flat, entire margin, surface: whitish, reverse: pale white, without pigmentation, covered with abundant, dense, downy aerial mycelium, reaching 64–65 mm diam. in 7 days. Colonies on OA: flat, entire margin, surface: hyaline to grayish sepia, reverse: smoky gray, without pigmentation, lacking aerial mycelium, with black or salmon spots in the center, reaching 58–63 mm diam. in 7 days.

Notes: *Colletotrichum philoxeroidis* formed a solitary clade near to the *C. magnum* species complex and the *C. orchidearum* species complex clades. Currently, the *C. magnum* species complex includes eight *Colletotrichum* species, and, except for *C. brevisporum* Noireung, Phouliv., L. Cai and K.D. Hyde, other species appear to be host-specific [43,46]. The *C. orchidearum* species complex also includes eight *Colletotrichum* species, but three species in this complex are very common and occur on many hosts, while the rest are less common, with some being host-specific and restricted to a specific region [43,46]. Based on a phylogenetic analysis, we suspected that *C. philoxeroidis* may not belong to the two species complexes and is a singleton species.

*Colletotrichum spicati* Z.F. Yu and H. Zheng, sp. nov. (Figure 8)

MycoBank Number: MB842297

Etymology: The species epithet is derived from the host plant *Myriophyllum spicatum*.

Holotype: YMF 1.04942, isolated as an endophyte from the stem of *Myriophyllum spicatum*, Caohai nature reserve, Weining county, Guizhou Province, China, August 2015, Z.F. Yu, preserved by lyophilization (a metabolically inactive state) in the State Key Laboratory for Conservation and Utilization of Bio-Resources in Yunnan. Ex-type culture: CGMCC3.18942.

Sexual morph not observed. Asexual morph on SNA. Vegetative hyphae: 1–8 μm diam., smooth-walled, hyaline, branched, septate. Chlamydospores not observed. Conidiophores formed directly on hyphae: smooth-walled, hyaline, branched, septate, up to 61 µm long. Conidiogenous cells: hyaline, smooth-walled, cylindrical or pyriform, 0.7–2.6 µm long. Conidia: hyaline, aseptate, smooth-walled, ellipsoidal to subglobose, sometimes cylindrical, the apex round and the base slightly pointed, sometimes with a prominent scar, containing 1–2 guttules, 10.9–15.7 µm × 5.0–8.2 µm. Appressoria: solitary, nearly spherical or cylindrical, smooth-walled, light brown to dark brown, 5.8–10.6 µm × 4.5–7.1 µm. 

Culture characteristics: Colonies on SNA: flat, entire margin, surface: hyaline, reverse: hyaline, without pigmentation, lacking aerial mycelium, reaching 55–60 mm diam. in 7 days. Colonies on PDA: entire margin, surface: white, reverse: pale white, without pigmentation, covered with abundant, loose aerial mycelium, reaching 64–70 mm diam. in 7 days. Colonies on MEA: flat, entire margin, surface: white, reverse: pale white, without pigmentation, covered with sparse, flocculent aerial mycelium, reaching 70 mm diam. in 7 days. Colonies on OA: flat, indistinct margin, surface: white, reverse: pale white, without pigmentation, dense aerial mycelium, abundant in the center and scarce in the margin, reaching 69–71 mm diam. in 7 days. 

Additional specimen examined: China, Yunnan Province, Chengjiang city, Fuxian Lake, isolated as an endophyte from leaf of *Potamogeton wrightii*, July 2014, Z.F. Yu, living culture F6.

Notes: Based on a multi-locus phylogenetic analysis, *Colletotrichum spicati* fell into the *C. boninense* species complex clade and close to the widespread species *C. karstii*. Morphologically, *C. spicati* shares similar morphological characteristics with other taxa in the *C. boninense* species complex, which have conidia with a prominent basal scar and conidiogenous cells with a rather prominent periclinal thickening [84]. Nevertheless, *C. spicati* is different from *C. karstii* by having smaller conidia; 10.9–15.7 µm × 5.0–8.2 µm vs. 12.5–19.5 µm × 6–8.5 μm [85].

*Colletotrichum tengchongense* Z.F. Yu and H. Zheng, sp. nov. (Figure 9)

MycoBank Number: MB842294

Etymology: The species epithet refers to Tengchong city, which is the site of collection of the type strain.

Holotype: YMF 1.04950, isolated as an endophyte from the root of *Isoetes sinensis*, Tengchong city, Yunnan Province, China, August 2014, Z.F. Yu, preserved by lyophilization (a metabolically inactive state) in the State Key Laboratory for Conservation and Utilization of Bio-Resources in Yunnan. Ex-type culture: CGMCC3.18950.

Sexual morph not observed. Asexual morph on SNA. Vegetative hyphae: 1–8 μm diam., hyaline, smooth-walled, septate, branched. Chlamydospores not observed. Conidiomata: acervular, setae and conidiophores formed from roundish brown cell. Setae: pale brown to dark brown, straight or curved, smooth-walled, 0–1 septate, 24–78 μm long, tip acute. Conidiophores: smooth-walled, hyaline to pale brown, branched, septate, cylindrical to clavate, up to 31 μm long. Conidiogenous cells: smooth-walled, hyaline to pale brown, monoblastic, integrated, terminal, lageniform or ampoule, 11.6–16.6 µm × 4.5–6.2 μm. Conidia: smooth-walled, hyaline, straight, the apex and base rounded, aseptate, 10.0–16.6 µm × 4.5–6.2 μm. Appressoria: solitary, navicular, lobate, irregular in outline or undulate margin, light brown to dark brown, 6.4–23.4 µm × 4.5–11.6 µm.

Culture characteristics: Colonies on SNA: flat, radial margin, surface: translucent, reverse: translucent, without pigmentation, covered with abundant, loose, flocculent aerial mycelium in the center, lacking aerial mycelium in the margin, reaching 71–74 mm diam. in 7 days. Colonies on PDA: entire margin, surface: white, reverse: pale white, without pigmentation, covered with abundant, dense, floccose aerial mycelium, reaching 50–60 mm diam. in 7 days. Colonies on MEA: indistinctly circular, entire margin, surface: whitish, reverse: pale white, without pigmentation, covered with abundant, downy aerial mycelium, reaching 44–48 mm diam. in 7 days. Colonies on OA: flat, circular, indistinct margin, surface: white, reverse: pale white, without pigmentation, covered with abundant, dense aerial mycelium in the center, with sparse aerial mycelium in the margin, reaching 78–80 mm diam. in 7 days.

Notes: Based on a multi-locus phylogenetic analysis, *Colletotrichum tengchongense* fell into the *C. gloeosporioides* species complex clade and near to the species *C. jiangxiense*. Morphologically, *C. tengchongense* shares similar morphological characteristics with other taxa in the gloeosporioides species complex, which have cylindrical conidia with rounded ends tapering slightly towards the base [86]. Furthermore, *C. tengchongense* is obviously different from *C. jiangxiense* by having lageniform or ampoule conidiogenous cells and smaller conidia (10.0–16.6 µm × 4.5–6.2 µm vs. 13–19 µm × 4–6 μm) [87].

*Colletotrichum vulgaris* Z.F. Yu and H. Zheng, sp. nov. (Figure 10)

MycoBank Number: MB842295

Etymology: The species epithet is derived from the host plant *Hippuris vulgaris*.

Holotype: YMF 1.04940, isolated as an endophyte from the leaf of *Hippuris vulgaris*, Baiyu county, Tibetan Autonomous Prefecture of Garzê, Sichuan Province, China, August 2015, Z.F. Yu, preserved by lyophilization (a metabolically inactive state) in the State Key Laboratory for Conservation and Utilization of Bio-Resources in Yunnan. Ex-type culture: CGMCC3.18940.

Sexual morph not observed. Asexual morph on SNA. Vegetative hyphae: 1–8 μm diam., smooth-walled, hyaline, branched, septate. Chlamydospores not observed. Conidiophores formed directly on hyphae: smooth-walled, hyaline, branched, septate, 5.0–66.0 µm × 2.0–4.0 µm. Conidiogenous cells: hyaline, smooth-walled, cylindrical to ampulliform, integrated, occasionally polyphialidic; phialides: discrete, 0.9–3.1 μm in length, apical opening 1.1–2.6 μm in diameter. Conidia: hyaline, aseptate, smooth-walled, variable in size and shape, thick-walled, curved, bent abruptly, tapering towards both ends, sometimes central part of conidia is usually nearly straight, containing 1–2 guttules, 16.6–23.7 (–32.7) µm × 3.7–5.2 μm. Appressoria: solitary, clavate to lobed, smooth-walled, one- or two-celled, light brown to dark brown, 7.0–23.5 µm × 4.6–8.9 µm. 

Culture characteristics: Colonies on SNA: flat, radial margin, surface: buff to cinnamon, reverse: buff to cinnamon, without pigmentation, covered with abundant, loose, lanose to cottony aerial mycelium, reaching 45–48 mm diam. in 7 days. Colonies on PDA: flat, radial margin, surface: white to pale gray, reverse: light olivaceous in the center and pale white in the margin, without pigmentation, covered with abundant, dense, floccose aerial mycelium, reaching 55–58 mm diam. in 7 days. Colonies on MEA: flat, inconspicuously circular, entire margin, surface: white to saffron, reverse: saffron in the center and pale white in the margin, without pigmentation, covered with abundant, loose aerial mycelium, reaching 60–65 mm diam. in 7 days. Colonies on OA: flat, indistinct margin, surface: white, reverse: pale gray, without pigmentation, covered with sparse, granular, aerial mycelium, reaching 40–50 mm diam. in 7 days. 

Notes: *Colletotrichum vulgaris* fell into the *C. gloeosporioides* species complex clade and is phylogenetically close to our new recognized species *C. tengchongense*. However, *C. vulgaris* is obviously different from *C. tengchongense* in morphology. For instance, *C. vulgaris* has curved conidia, while the conidia of *C. tengchongense* are straight. In addition, the ITS sequence of *C. vulgaris* just has 92% identity with it.

*Colletotrichum wuxuhaiense* Z.F. Yu and H. Zheng, sp. nov. (Figure 11)

MycoBank Number: MB842296

Etymology: The species epithet refers to Wuxuhai scenic area, which is the site of collection of the type strain.

Holotype: YMF 1.04951, isolated as an endophyte from the root of *Potamogeton crispus*, Wuxuhai scenic area, Jiulong county, Tibetan Autonomous Prefecture of Garzê, Sichuan Province, China, August 2015, Z.F. Yu, preserved by lyophilization (a metabolically inactive state) in the State Key Laboratory for Conservation and Utilization of Bio-Resources in Yunnan. Ex-type culture: CGMCC3.18951.

Sexual morph developed on SNA. Ascomata perithecia: solitary or in clusters, superficial or partly immersed in medium, globose to near globose, ostiolate, black, glabrous, 138.8–153.4 µm × 120.9–127.4 μm. Asci: hyaline, cylindrical to clavate, 50.7–68.8 µm × 8.4–12.4 μm, eight-spored. Ascospores: smooth-walled, hyaline, aseptate, fusiform, slightly curved, both sides rounded or one side pointed, contents: granular, 13.4–17.2 µm × 4.1–6.6 μm. Asexual morph on SNA. Vegetative hyphae: 1–8 μm diam., hyaline, smooth-walled, septate, branched. Chlamydospores not observed. Conidiophore formed directly on hyphae: smooth-walled, hyaline, branched, septate, cylindrical to clavate, up to 31 μm long. Conidiogenous cells: smooth-walled, hyaline, monoblastic, integrated, terminal, cylindrical to clavate, 11.6–15.8 µm × 4.7–8.9 μm. Conidia: smooth-walled, hyaline, aseptate, oval, the apex and base rounded, sometimes the base slightly pointed with a prominent scar, 10.3–15.8 µm × 4.7–8.9 μm. Appressoria: solitary, navicular, lobate, irregular in outline or undulate margin, light brown to dark brown, 5.9–18.9 µm × 4.6–7.3 µm. 

Culture characteristics: Colonies on SNA: flat, entire margin, surface: translucent, reverse: translucent, without pigmentation, lacking aerial mycelium, reaching 45–54 mm diam. in 7 days. Colonies on PDA: entire margin, surface: white, reverse: pale white, without pigmentation, covered with abundant, dense, floccose aerial mycelium, reaching 50–63 mm diam. in 7 days. Colonies on MEA: flat, circular, radial margin, surface: translucent to white, reverse: translucent to pale white, without pigmentation, covered with sparse, loose aerial mycelium, reaching 45–54 mm diam. in 7 days. Colonies on OA: flat, circular, indistinct margin, surface: translucent, reverse: translucent, without pigmentation, covered with sparse, minute aerial mycelium in the center, lacking aerial mycelium in the margin, reaching 64–69 mm diam. in 7 days.

Additional specimen examined: China, Yunnan Province, Chengjiang city, Fuxian Lake, isolated as an endophyte from the root of *Potamogeton pectinatus*, July 2014, Z.F. Yu, living culture F34.

Notes: *Colletotrichum wuxuhaiense* fell into the *C. boninense* species complex clade and was also close to *C. karstii* in our phylogenetic tree. Morphologically, *C. wuxuhaiense* shares similar morphological characteristics with other taxa in the *C. boninense* species complex, which have conidia with a prominent basal scar and conidiogenous cells with a rather prominent periclinal thickening [84]. However, *C. wuxuhaiense* is distinguished from *C. karstii* by having shorter conidia; 10.3–15.8 µm × 4.7–8.9 µm vs. 12.5–19.5 µm × 6–8.5 μm [84].

### 3.3. Isolation Frequency of Colletotrichum Species in Different Sampling Sites

During the investigation of the diversity of endophytic fungi in aquatic plants in southwest China by Zheng et al. [65], a total of 1689 endophytic fungi were isolated, but there were only 66 strains belonging to *Colletotrichum* species. Their study indicated that *Colletotrichum* is not a dominant genus in aquatic plants in southwest China. However, *Colletotrichum* species were isolated from 18 sampling sites; this proves again that *Colletotrichum* species are widely distributed. The isolation frequencies and numbers of *Colletotrichum* species at different sites are shown in Table 1. The total isolation frequencies in root, stem, and leaf tissues were 21.0%, 46.8%, and 32.3%, respectively. This indicates that *Colletotrichum* species are distributed in the root, stem, and leaf tissues of aquatic plants, but are most abundant in stem tissues. Of these, the most isolates were obtained in the Baiyu sampling site, followed by Dian Lake and Litang. 

### 3.4. Pathogenicity Assay

Pathogenicity was tested for our newly isolated taxa, but *Colletotrichum demersi* and *C. casaense* were not tested for pathogenicity and virulence because the concentration of their spore suspensions was too low. The results of the pathogenicity test on five fruits are shown in Table 2. According to the number of fruits in which disease could be caused, the aggressiveness was classified as strongly virulent (>70%), moderately virulent (≥30% to ≤70%), or weakly virulent (<30%). We can see from Table 2 that the aggressiveness of the tested isolates differed significantly. For instance, *C. garzense* could only cause disease for tomatoes, but the other seven species could cause disease for five fruits. In addition, we calculated the virulence levels based on the lesion size and the number of spores (Table 3). As shown in Table 3, the virulence levels of *Colletotrichum* species in five fruits also had significant differences. For example, the species *C. wuxuhaiense* had a strong virulence level in four fruits, including grape, tangerine, tomato, and blueberry, but had weak virulence in strawberry. In all, the results for the virulence level were consistent with the pathogenicity. 

## 4. Discussion

A previous study by Zheng et al. [65] investigated the diversity of endophytic fungi in aquatic plants in southwest China and obtained 66 isolates belonging to *Colletotrichum* species. In this study, we continued to identify these endophytic *Colletotrichum* isolates at the species level. Based on morphology and a concatenated five-gene phylogenetic analysis, 26 endophytic *Colletotrichum* isolates were assigned to different species complexes, and further study identified these isolates as 13 species, including three known species—*C. fructicola*, *C. jiangxiense*, and *C. karstii*—and ten new species—*C. baiyuense*, *C. casaense*, *C. demersi*, *C. dianense*, *C. garzense*, *C. philoxeroidis*, *C. spicati*, *C. tengchongense*, *C. vulgaris*, and *C. wuxuhaiense*. Two species complexes, the *Colletotrichum boninense* species complex and *Colletotrichum gloeosporioides* species complex, were found to be associated with aquatic plants. In order to prove Koch’s postulates, pathogenicity tests on five common fruits revealed a broad diversity in pathogenicity and aggressiveness among these *Colletotrichum* species.

The name *Colletotrichum gloeosporioides* was first proposed by Penzig [88] based on *Vermicularia gloeosporioides*, and the type strain was collected from *Citrus* in Italy. Now, the *Colletotrichum gloeosporioides* species complex is one of the largest complexes within the genus, comprising 52 species [30,31,46,47,89]. Species in this complex are mainly plant pathogens [49,86,90,91,92,93], but some species are endophytes, such as those isolated from *Camellia* spp. [87], *Merremia umbellata* [94], and *Pennisetum purpureum* [95]. Species in the gloeosporioides complex are mainly characterized by species with cylindrical conidia with rounded ends that taper slightly towards the base [86]. Previous studies based on the multi-gene phylogeny showed that the species complex includes two subclades [46,47,86], namely, kahawae and musae. In this study, a combined multi-gene analysis of ITS, GAPDH, CHS-1, ACT, and TUB2 sequence data is shown in Figure 1. Our results also recognized the two subclades within the species complex, and our seven endophytic isolates were located within the *Colletotrichum gloeosporioides* species complex, including two known and two new species. The two known species are *C. fructicola* Prihastuti, L. Cai and K.D. Hyde and *C. jiangxiense* F. Liu and L. Cai. *C. fructicola* was first reported in coffee berries in Thailand [94]. This species has a wide host range and geographical distribution [31]. This species also occurs as an endophyte. We isolated two strains of *C. fructicola* from aquatic plants in Yunnan province, and this confirms its widespread distribution again. *C. jiangxiense* was first reported from *Camellia sinensis* in Jiangxi Province, China, and is a pathogen as well as an endophyte [87]. Three of our endophytic isolates were identified as *C. jiangxiense*; of these, two were isolated from *Isoetes sinensis* in Sichuan province and one from *Potamogeton pectinatus* in Yunnan province. Therefore, this study expands the habitat of *C. jiangxiense*. In addition, two new taxa were recognized in the *Colletotrichum gloeosporioides* species complex, namely, *C. vulgaris* and *C. tengchongense*.

*Colletotrichum boninense* was first described in *Crinum asiaticum* var. *sinicum* collected in the Bonin Islands, Japan [96]. Later, this species was found to occur in a range of hosts. The *Colletotrichum boninense* species complex is defined as a collective of *C. boninense* and 25 closely related species. Typical characteristics of species in this complex are the conidia that have a prominent basal scar, as well as the conidiogenous cells with a rather prominent periclinal thickening that sometimes extends to form a new conidiogenous locus [84]. Species in the boninense complex have been isolated as pathogens or endophytes [84]. In this study, we obtained five endophytic isolates belonging to this complex from aquatic plants, including one known and two new species. *C. karstii* is the most common and geographically diverse species in the boninense complex [84,85]. We first obtained the species isolate from the host plant *Potamogeton wrightii*. Most importantly, we found two new taxa in this species complex in aquatic plants, namely, *C. spicati* and *C. wuxuhaiense*. 

In this study, pathogenicity tests on five common fruits were conducted using eight new *Colletotrichum* species isolates, which are *C. baiyuense*, *C. dianense*, *C. garzense*, *C. philoxeroidis*, *C. spicati*, *C. tengchongense*, *C. vulgaris*, and *C. wuxuhaiense*. Seven of them showed strong pathogenicity in the five fruits. Furthermore, all tested *Colletotrichum* species showed significantly different virulence levels in these five fruits, and the results were consistent with those of the pathogenicity test. Therefore, our studies revealed a broad diversity in pathogenicity and aggressiveness among our newly recognized *Colletotrichum* species. Of course, symptoms may be easily affected by many factors, such as the cultivar and fruit physiological state. Thus, the pathogenicity test results from this study may not reflect the full aggressiveness potential of the isolates examined. Notably, all test strains were isolated from healthy tissues of aquatic plants. This indicates that our recognized species could switch their endophytic lifestyle to become plant pathogens. Our results also provide convincing evidence for the hypothesis that endophytes can be latent pathogens [97,98].

It is well known that *Colletotrichum* species have a wide host range and geographic distribution [31,39,46,47]. In fact, *Colletotrichum* species are also some of the most common endophytes. Many *Colletotrichum* species could exist in living plant tissues without causing disease [31,44,99]. Of course, some endophytic *Colletotrichum* species have mutualistic interactions with their plant hosts, including providing disease resistance, drought tolerance, and growth enhancements [100,101]. Currently, the majority of *Colletotrichum* species in the three species complexes—the boninense, gloeosporioides, and graminicola-caudatum complexes—are endophytes [31,46,84,86]. Previous studies investigating endophytic *Colletotrichum* species in plants also found that most isolates belong to the boninense and gloeosporioides complexes [44,102]. In our 26 identified isolates, seven and five isolates, respectively, belonged to the gloeosporioides and boninense complexes, and twelve isolates were clustered as a large group near the graminicola-caudatum complex.

Over the last decade, our knowledge of fungi and their relationships with plant hosts has increased exponentially due to the progress in bioinformatics and molecular phylogenetics. *Colletotrichum* species are important phytopathogens in many economically important hosts around the world; therefore, identification of the species associated with hosts, as well as their lifestyles, is important [31]. This study conducted a large-scale investigation of *Colletotrichum* spp. associated with aquatic plants in southwest China and provides morphological, molecular, and biological characterizations of these endophytic *Colletotrichum* isolates. We finally recognized ten new *Colletotrichum* species, and this is the first report of *C. jiangxiense* in two aquatic plants. This study not only enhances our understanding of the diversity of *Colletotrichum* species associated with aquatic plants, but also enriches knowledge of the host diversity of *Colletotrichum* species.

## Figures and Tables

**Figure 1 jof-08-00087-f001:**
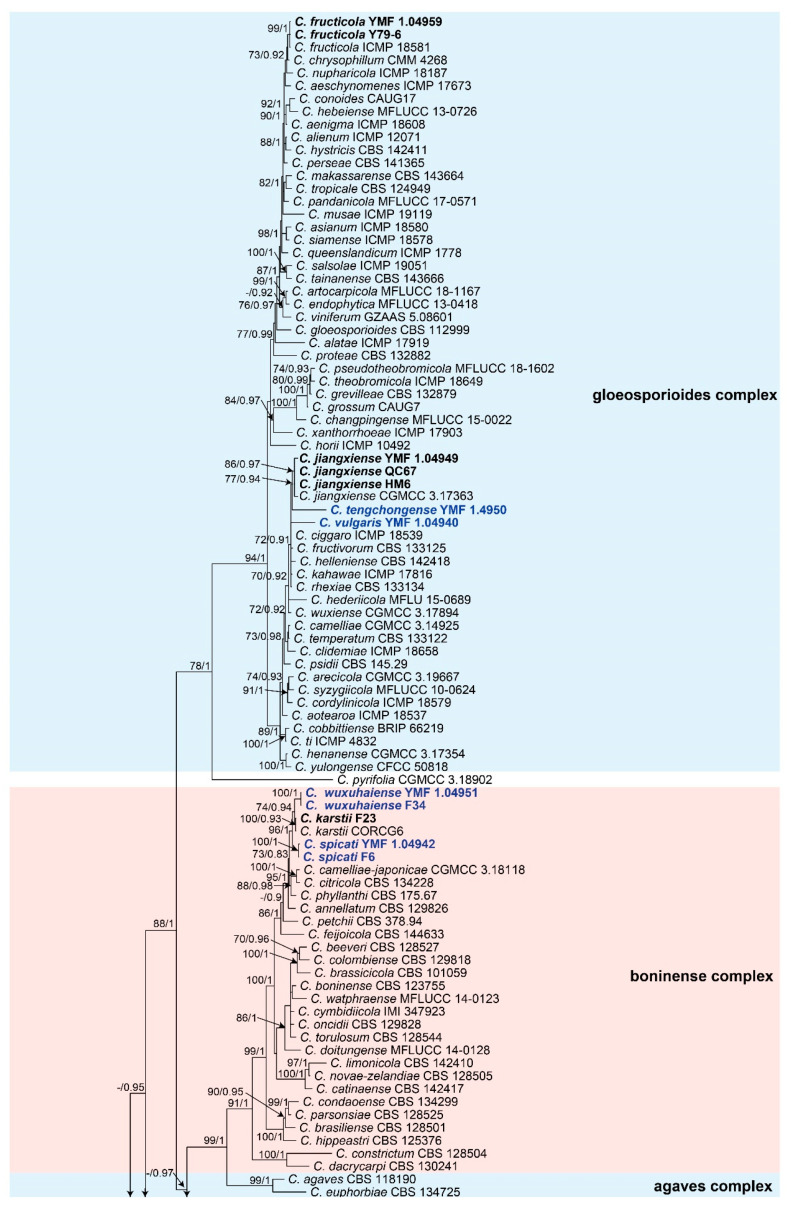
Phylogenetic tree generated by the Bayesian inference analysis using combined ITS, GAPDH, CHS-1, ACT, and TUB2 sequence data. Bootstrap values of ≥70% (**top**) and Bayesian posterior probability values of ≥90% (**bottom**) are indicated at the nodes (MLBP/BIBP). *Monilochaetes infuscans* CBS 86996 was used as the outgroup in this tree. The known species we isolated are bolded in black and new taxa are bolded in blue.

**Figure 2 jof-08-00087-f002:**
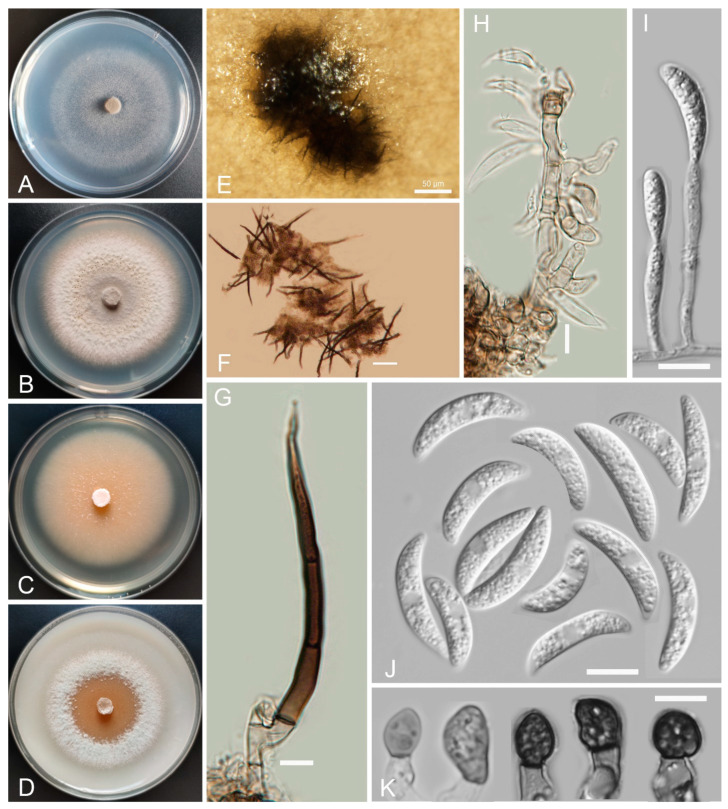
*Colletotrichum baiyuense* (YMF 1.04941). (**A**–**D**) Cultures on (**A**) SNA, (**B**) PDA, (**C**) MEA, and (**D**) OA for 7 days at 25 °C. (**E**) Conidiomata. (**F**,**G**) Seta. (**H**,**I**) Conidiophores and conidia. (**J**) Conidia. (**K**) Appressoria. Scale bars: (**E**) = 50 μm, (**F**) = 20 μm, (**G**,**K**) = 10 μm.

**Figure 3 jof-08-00087-f003:**
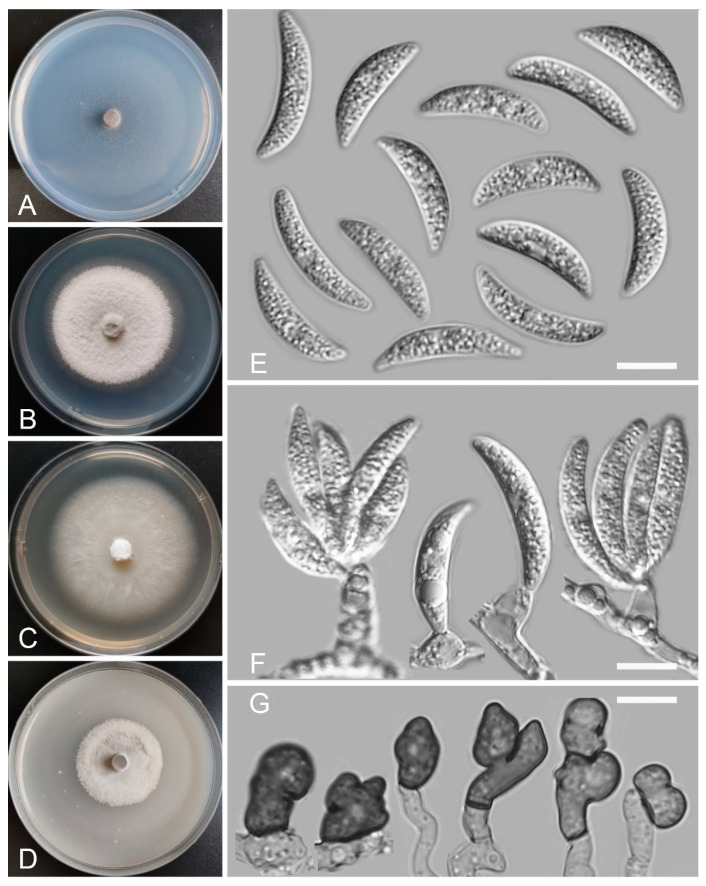
*Colletotrichum casaense* (YMF 1.04947). (**A**–**D**) Cultures on (**A**) SNA, (**B**) PDA, (**C**) MEA, and (**D**) OA for 7 days at 25 °C. (**E**) Conidia. (**F**) Conidiophores and conidia. (**G**) Appressoria. Scale bars: (**E**–**G**) = 10 μm.

**Figure 4 jof-08-00087-f004:**
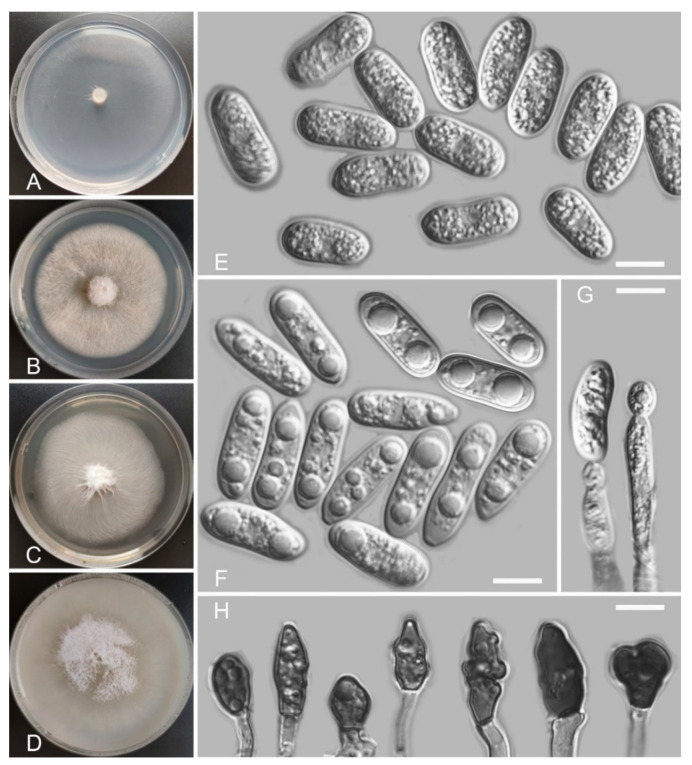
*Colletotrichum demersi* (YMF 1.04946). (**A**–**D**) Cultures on (**A**) SNA, (**B**) PDA, (**C**) MEA, and (**D**) OA for 7 days at 25 °C. (**E**,**F**) Conidia. (**G**) Conidiophores and conidia. (**H**) Appressoria. Scale bars: (**E**–**H**) = 10 μm.

**Figure 5 jof-08-00087-f005:**
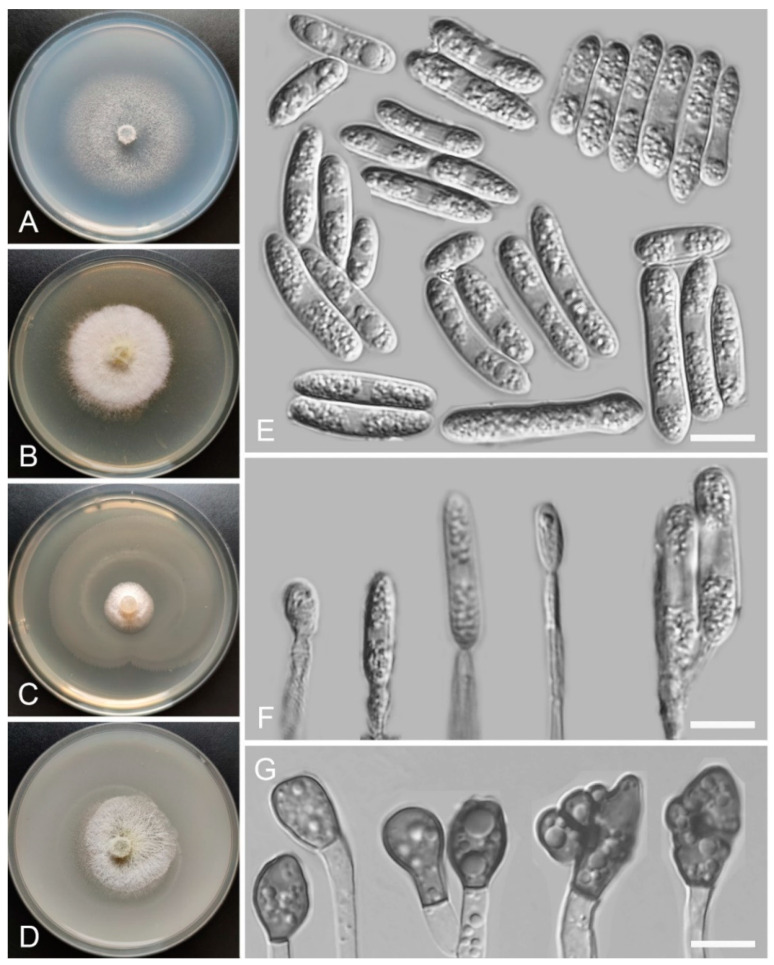
*Colletotrichum dianense* (YMF 1.04943). (**A**–**D**) Cultures on (**A**) SNA, (**B**) PDA, (**C**) MEA, and (**D**) OA for 7 days at 25 °C. (**E**) Conidia. (**F**) Conidiophores and conidia. (**G**) Appressoria. Scale bars: (**E**–**G**) = 10 μm.

**Figure 6 jof-08-00087-f006:**
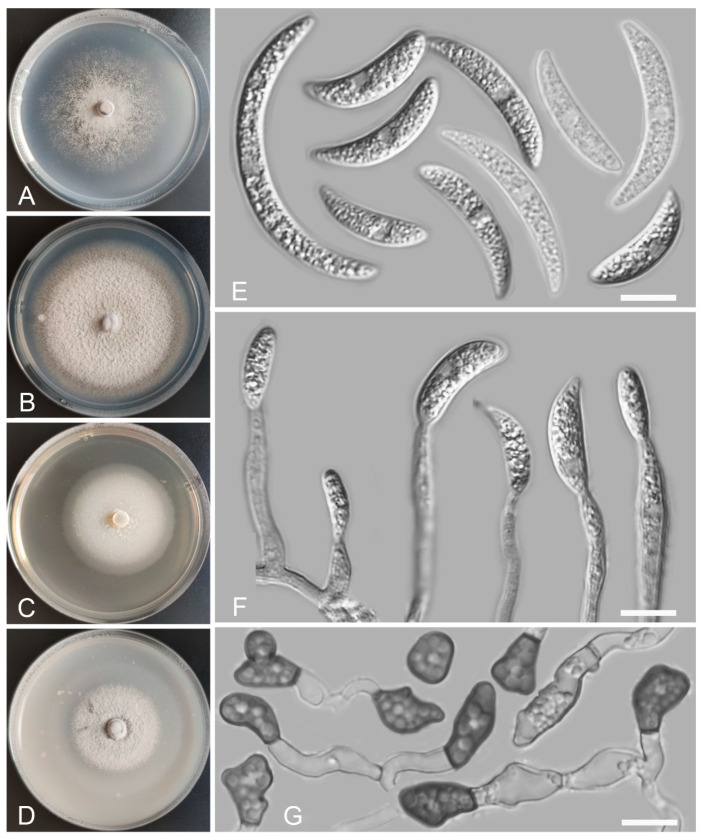
*Colletotrichum garzense* (YMF 1.04948). (**A**–**D**) Cultures on (**A**) SNA, (**B**) PDA, (**C**) MEA, and (**D**) OA for 7 days at 25 °C. (**E**) Conidia. (**F**) Conidiophores and conidia. (**G**) Appressoria. Scale bars: (**E**–**G**) = 10 μm.

**Figure 7 jof-08-00087-f007:**
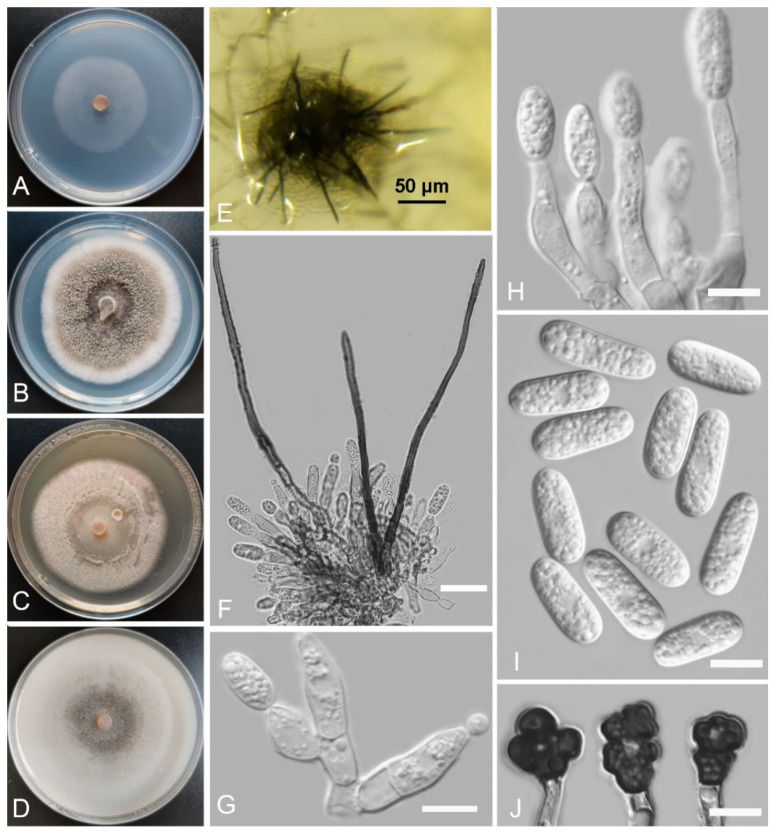
*Colletotrichum philoxeroidis* (YMF 1.04945). (**A**–**D**) Cultures on (**A**) SNA, (**B**) PDA, (**C**) MEA, and (**D**) OA for 7 days at 25 °C. (**E**) Conidiomata. (**F**) Seta and conidiophores. (**G**,**H**) Conidiophores and conidia. (**I**) Conidia. (**J**) Appressoria. Scale bars: (**E**) = 50 μm, (**F**) = 20 μm, (**G**–**J**) = 10 μm.

**Figure 8 jof-08-00087-f008:**
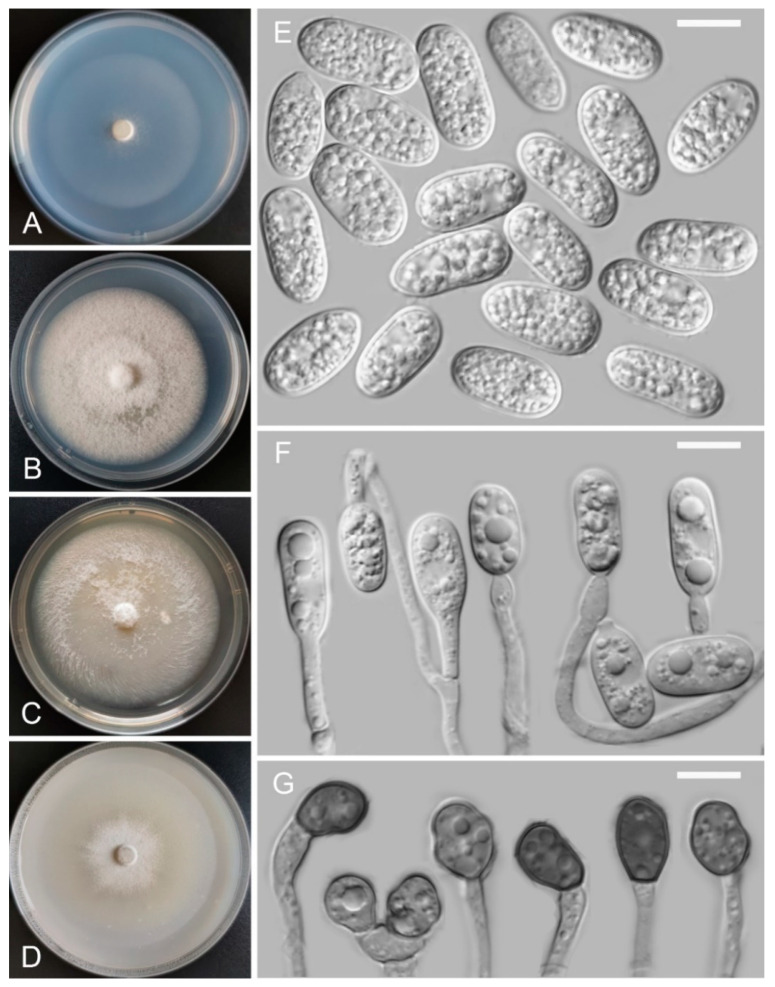
*Colletotrichum spicati* (YMF 1.04942). (**A**–**D**) Cultures on (**A**) SNA, (**B**) PDA, (**C**) MEA, and (**D**) OA for 7 days at 25 °C. (**E**) Conidia. (**F**) Conidiophores and conidia. (**G**) Appressoria. Scale bars: (**E**–**G**) = 10 μm.

**Figure 9 jof-08-00087-f009:**
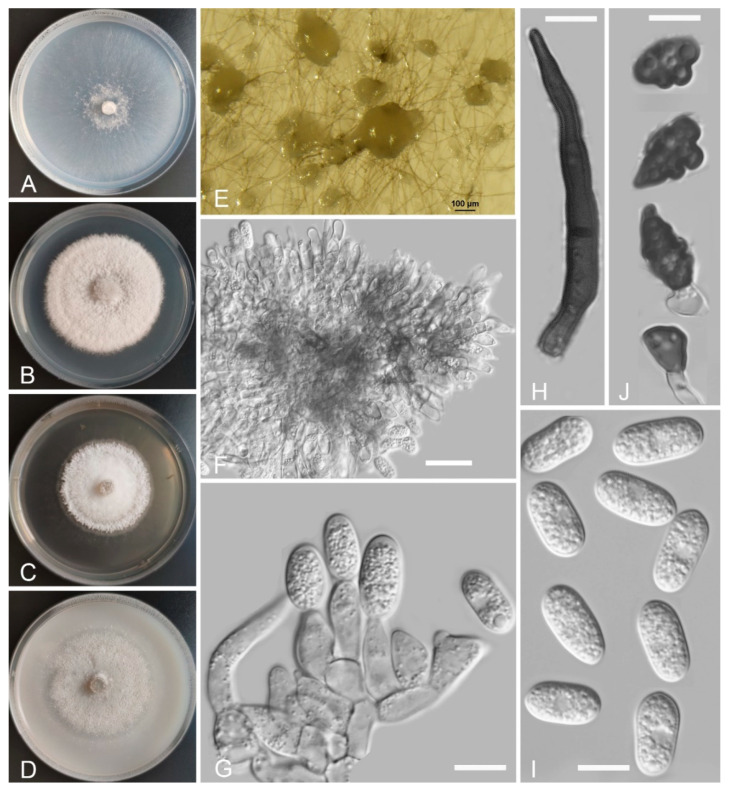
*Colletotrichum tengchongense* (YMF 1.04950). (**A**–**D**) Cultures on (**A**) SNA, (**B**) PDA, (**C**) MEA, and (**D**) OA for 7 days at 25 °C. (**E**) Conidiomata. (**F**) Conidiophores. (**G**) Conidiophores and conidia. (**H**) Seta. (**I**) Conidia. (**J**) Appressoria. Scale bars: (**E**) = 100 μm, (**F**) = 20 μm, (**G**–**J**) = 10 μm.

**Figure 10 jof-08-00087-f010:**
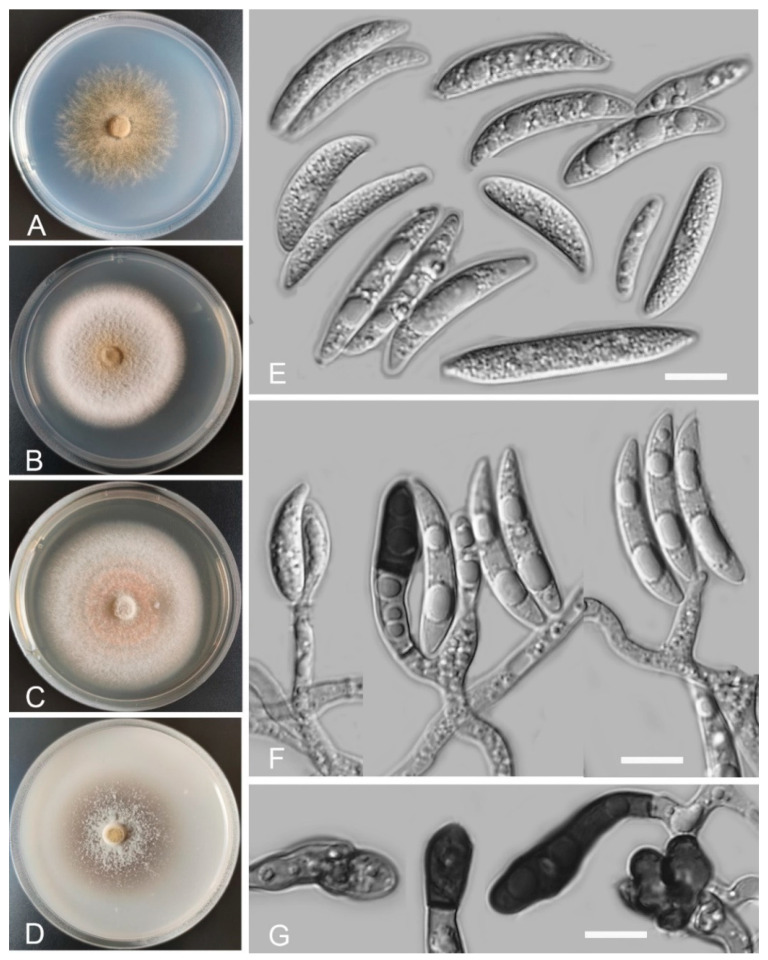
*Colletotrichum vulgaris* (YMF 1.04940). (**A**–**D**) Cultures on (**A**) SNA, (**B**) PDA, (**C**) MEA, and (**D**) OA for 7 days at 25 °C. (**E**) Conidia. (**F**) Appressoria, Conidiophores and conidia. (**G**) Appressoria. Scale bars: (**E**–**G**) = 10 μm.

**Figure 11 jof-08-00087-f011:**
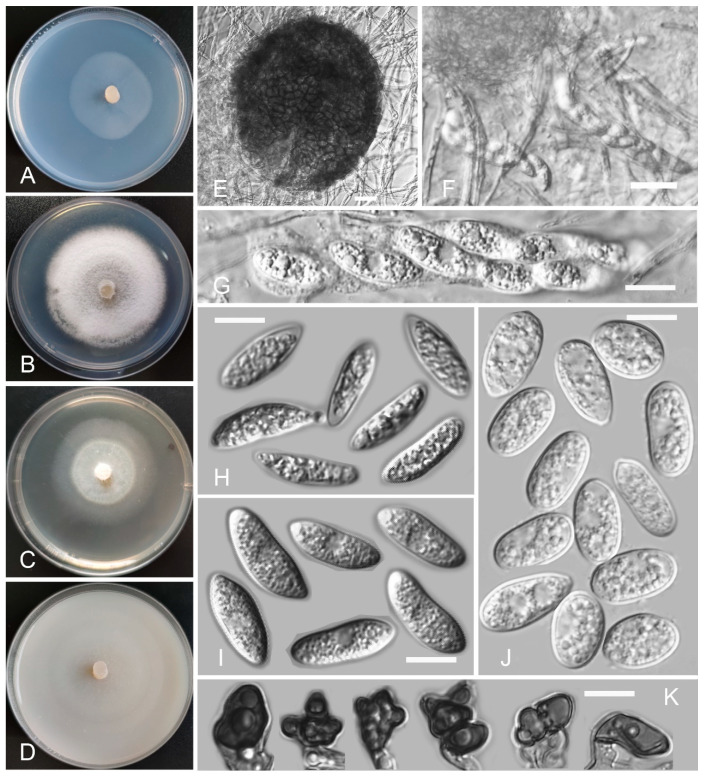
*Colletotrichum wuxuhaiense* (YMF 1.04951). (**A**–**D**) Cultures on (**A**) SNA, (**B**) PDA, (**C**) MEA, and (**D**) OA for 7 days at 25 °C. (**E**) Ascomata. (**F**,**G**) Asci. (**H**,**I**) Ascospores. (**J**) Conidia. (**K**) Appressoria. Scale bars: (**E**,**F**) = 20 μm, (**G**–**K**) = 10 μm.

**Table 1 jof-08-00087-t001:** Isolation frequency of endophytic *Colletotrichum* species from aquatic plants in different sites.

Provinces	Sampling Sites	Isolation Frequency				Number of Isolates
		Root	Stem	Leaf	Total	
Yunnan	Dian lake	33.3%	44.4%	22.2%	15.2%	10
	Fuxian lake	33.3%	16.7%	50.0%	9.1%	6
	Erhai	33.3%	33.3%	33.3%	4.6%	3
	Tengchong	100.0%	0.0%	0.0%	3.0%	2
	Yangzonghai	100.0%	0.0%	0.0%	3.0%	2
	Huamajie	0.0%	100.0%	0.0%	1.5%	1
	Jianhu	0.0%	0.0%	100.0%	1.5%	1
	Qincaitang	0.0%	0.0%	100.0%	1.5%	1
	Xianggelila	0.0%	100.0%	0.0%	1.5%	1
	Yila	0.0%	100.0%	0.0%	1.5%	1
Sichuan	Baiyu	5.3%	57.9%	36.8%	28.8%	19
	Litang	28.6%	57.1%	14.3%	10.6%	7
	Casa lake	0.0%	100.0%	0.0%	4.6%	3
	Qionghai	0.0%	0.0%	100.0%	3.0%	2
	Wuxuhai	100.0%	0.0%	0.0%	1.5%	1
Guizhou	Hongfeng lake	0.0%	25.0%	75.0%	6.1%	4
	Caohai	0.0%	100.0%	0.0%	1.5%	1
	Huaxi	100.0%	0.0%	0.0%	1.5%	1
Total		21.0%	46.8%	32.3%		

**Table 2 jof-08-00087-t002:** Pathogenicity testing for eight new *Colletotrichum* species.

Species	Days	Lesion Diameter (mm)					Incidence Rate (%)						Pathogenicity
		Strawberry	Grape	Tangerine	Tomato	Blueberry	Strawberry	Grape	Tangerine	Tomato	Blueberry	Total	
*C. vulgaris*	7	24.5	2.5	3.5	7.8	2.0	100%	100%	100%	100%	100%	100%	Strong
	14	whole cover	3.7	5.5	8.0	3.5	100%	100%	100%	100%	100%	100%	
*C. baiyuense*	7	5.0	0.0	5.5	3.7	2.5	100%	0%	100%	100%	100%	80%	Strong
	14	30.0	0.0	6.0	11.0	3.0	100%	0%	100%	100%	100%	80%	
*C. spicati*	7	12.0	6.7	12.0	21.0	8.3	100%	100%	100%	100%	100%	100%	Strong
	14	whole cover	10.0	whole cover	26.0	13.3	100%	100%	100%	100%	100%	100%	
*C. dianense*	7	13.5	4.3	5.5	16.5	4.7	100%	100%	100%	100%	100%	100%	Strong
	14	18.0	9.0	7.5	whole cover	12.0	100%	100%	100%	100%	100%	100%	
*C. philoxeroidis*	7	5.3	9.0	4.5	8.5	3.3	100%	100%	100%	100%	100%	100%	Strong
	14	6.0	12.0	5.5	16.5	4.0	100%	100%	100%	100%	100%	100%	
*C. garzense*	7	0.0	0.0	0.0	5.0	0.0	0%	0%	0%	100%	0%	20%	Weak
	14	0.0	0.0	0.0	6.0	0.0	0%	0%	0%	100%	0%	20%	
*C. tengchongense*	7	8.5	5.5	8.0	16.5	3.3	100%	100%	100%	100%	100%	100%	Strong
	14	33.0	11.0	16.0	whole cover	7.3	100%	100%	100%	100%	100%	100%	
*C. wuxuhaiense*	7	7.0	13.3	5.5	17.0	6.3	100%	100%	100%	100%	100%	100%	Strong
	14	18.0	19.0	6.0	31.0	15.0	100%	100%	100%	100%	100%	100%	

**Table 3 jof-08-00087-t003:** Virulence levels of eight *Colletotrichum* species in five fruits.

Species	Strawberry	Grape	Tangerine	Tomato	Blueberry
*C. vulgaris*	6	5	5	5	5
*C. baiyuense*	3	0	5	5	4
*C. spicati*	6	5	6	5	5
*C. dianense*	5	3	3	6	5
*C. philoxeroidis*	5	5	4	5	5
*C. garzense*	0	0	0	5	0
*C. tengchongense*	5	5	3	6	5
*C. wuxuhaiense*	3	5	5	5	5

## Data Availability

All sequence data are available in NCBI GenBank following the accession numbers in the manuscript.

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
