# Peer review of "Endophytic Colletotrichum Species from Aquatic Plants in Southwest China"

_jof, 2022, doi:10.3390/jof8010087_

Round 1
Reviewer 1 Report
Please see the attached file including comments

Author Response
Dear Reviewer:
Thank you for your comments concerning our manuscript. Those comments are all valuable and very helpful for revising and improving our paper. We have made correction according your suggestions. Revised portion have done in the paper.
Point: This is an interesting study and the authors have investigated endophytic Colletotrichum species occurring within aquatic plants that are little known because most studies have investigated the diversity and pathogenicity of these species in common ornamentals, fruits, and vegetables. I think this work is considered a pioneer in this subject. The paper is generally well written and structured. The paper is also scientifically good. Overall, the information presented here represents valuable information regarding the endophytic Colletotrichum species occurring within aquatic plants. This paper will be helpful for scientists in the field of mycology
However, there are many language mistakes; I corrected many of them as listed below:
Response: Thank you so much for your corrections, we have amended these in manuscript. In addition, we have carefully checked the manuscript again and made changes, but these changes will not influence the content and framework of the paper. Special thanks to you for your good comments.
Reviewer 2 Report
I find the article submitted for review interesting. It is written correctly and understandably. It complements the previously published article (https://doi.org/10.3389/ffunb.2021.69254), to which the authors refer in the methodology. The title is consistent with the content and methodology. The introduction is clear and provides a concrete background for the research. However, I lack a clearly defined goal and research assumptions. The aim of the study is presented as a research result rather than an assumption. I know this article is a continuation of research that was previously published, but there is an inaccuracy here because the publication by Zheng et al. 2021, it was indicated that the research was conducted in 2015, and the article I am reviewing mentions 2014 and 2015. Please make a correction. There is no space on line 102. The results are presented clearly, the figures are clear. The description of the species is correct. The results are substantively discussed, and the cited literature is very extensive. After taking into account a minor correction, I recommend the article to be printed without re-reviewing it.
Author Response
Dear Reviewer:
Thank you for your comments concerning our manuscript. Those comments are all valuable and very helpful for revising and improving our paper. We have made correction according your suggestions. Revised portion have done in the paper.
Point: I find the article submitted for review interesting. It is written correctly and understandably. It complements the previously published article (https://doi.org/10.3389/ffunb.2021.69254), to which the authors refer in the methodology. The title is consistent with the content and methodology. The introduction is clear and provides a concrete background for the research. However, I lack a clearly defined goal and research assumptions. The aim of the study is presented as a research result rather than an assumption. I know this article is a continuation of research that was previously published, but there is an inaccuracy here because the publication by Zheng et al. 2021, it was indicated that the research was conducted in 2015, and the article I am reviewing mentions 2014 and 2015. Please make a correction. There is no space on line 102. The results are presented clearly, the figures are clear. The description of the species is correct. The results are substantively discussed, and the cited literature is very extensive. After taking into account a minor correction, I recommend the article to be printed without re-reviewing it.
Response: Thank you so much for your suggestion. About the sampling data, I have checked again. We found that there are some errors about data in the publication by Zheng et al. 2021. We collected samples in Yunnan Province between July and September 2014, in Guizhou province in July 2015, and in Sichuan province between August – September 2015. Special thanks to you for your good comments.
Reviewer 3 Report
The manuscript is acceptable for publication after minor revision. A few points need to be addressed:
- Branch lengths of C. dianense, C. tengchongense, C. vulgaris are extremely short (indicating less distinguishable) or strangely long. A stronger argument for their novelty should be given. For example, the percentage similarity or number of bp difference in each should be given, especially when new species are introduced based on single specimen. While for those species sitting on a strangely long branch need to be checked for sequence quality, just to make sure they were sequenced with no error.
- C. demersi is probably sharing equal or similar phylogenetic position (may or may not) with two other endophyte C. bambusicola C.L. Hou & Q.T. Wang or C. guangxiense C.L. Hou & Q.T. Wang (Mycologia 2021, 113). Now all people are highly dependent on a phylogenetic tree to distinguish new species, but the premise is that all similar species are included in the tree.
- Why C. casaense, C. garzense, C. baiyuense do not belong to Graminicola-caudatum complex?
Author Response
Dear Reviewer:
Thank you for your comments concerning our manuscript. Those comments are all valuable and very helpful for revising and improving our paper.
Point 1: Branch lengths of C. dianense, C. tengchongense, C. vulgaris are extremely short (indicating less distinguishable) or strangely long. A stronger argument for their novelty should be given. For example, the percentage similarity or number of bp difference in each should be given, especially when new species are introduced based on single specimen. While for those species sitting on a strangely long branch need to be checked for sequence quality, just to make sure they were sequenced with no error.
Response: It is really true as you said species sitting on a strangely long branch need to be checked for sequence quality. In preliminarily phylogenetic tree trying four, five or six loci, these results showed that the three species all formed solitary clades easily distinguishing from other species. Considering the Reviewer’s suggestion, we added comparisons in the notes section of manuscript.
Point 2: C. demersi is probably sharing equal or similar phylogenetic position (may or may not) with two other endophyte C. bambusicola C.L. Hou & Q.T. Wang or C. guangxiense C.L. Hou & Q.T. Wang (Mycologia 2021, 113). Now all people are highly dependent on a phylogenetic tree to distinguish new species, but the premise is that all similar species are included in the tree.
Response: We are very sorry for our negligence of not adding the two species in our phylogenetic analysis. In this study, we constructed the phylogenetic tree based on the recent studies by Jayawardena et al. 2021, 2020. If now the two species are added for reconstructing phylogenetic tree, it will need relatively long time to revise our manuscript. By comparing C. demersi with C. bambusicola and C. guangxiense in morphology according to the article by Wang et al. (2021), we found that C. demersi is easily distinguished from them by lacking of conidiomata and setae. Moreover, the conidia of C. demersi is cylindrical, the apex rounded and base acute, sometimes with prominent scar, 12.9–20.9 × 4.8–6.9 μm while C. bambusicola is cylindrical to clavate, 13–17 × 4–5 μm and C. guangxiense is cylindrical to ellipsoidal, both ends rounded, 11–14 × 6.5–7.5 μm. Therefore, C. demersi can be described as a new species distinguishing from C. bambusicola and C. guangxiense.
Point 3: Why C. casaense, C. garzense, C. baiyuense do not belong to Graminicola-caudatum complex?
Response: C. casaense, C. garzense, C. baiyuense and C. demersi were phylogenetically closed to the Graminicola-caudatum complex clade, but the first three have curved conidia while C. demersi has cylindrical conidia. Our preliminary tree (No displaying in manuscript) included 25 species of the Graminicola-caudatum complex revealed that the four species clustered together and significantly distinct from Graminicola-caudatum complex clade. Therefore, we think the three species may not belong to the Graminicola-caudatum complex.
We appreciate for Reviewers’ warm work earnestly, and hope that the correction will meet with approval.
Once again, thank you very much for your comments and suggestions.